# Translating Urban Walkability Initiatives for Older Adults in Rural and Under-Resourced Communities

**DOI:** 10.3390/ijerph16173041

**Published:** 2019-08-22

**Authors:** Alexandra Klann, Linh Vu, Mollie Ewing, Mark Fenton, Rachele Pojednic

**Affiliations:** 1Department of Nutrition, Simmons University, Boston, MA 02115, USA; 2Children and Family Services Corporation, Vincennes, IN 47951, USA; 3Friedman School of Nutrition Science and Policy, Tufts University, Boston, MA 02111, USA; 4Harvard Medical School, Institute of Lifestyle Medicine, Boston, MA 02115, USA

**Keywords:** walkability, parks, recreation, physical activity, exercise, older adults, infrastructure, community gardens

## Abstract

The built environment can promote physical activity in older adults by increasing neighborhood walkability. While efforts to increase walkability are common in urban communities, there is limited data related to effective implementation in rural communities. This is problematic, as older adults make up a significant portion of rural inhabitants and exhibit lower levels of physical activity. Translating lessons from urban strategies may be necessary to address this disparity. This review examines best practices from urban initiatives that can be implemented in rural, resource-limited communities. The review of the literature revealed that simple, built environment approaches to increase walkability include microscale and pop-up infrastructure, municipal parks, and community gardens, which can also increase physical activity in neighborhoods for urban older adults. These simple and cost-effective strategies suggest great potential for rural communities.

## 1. Introduction

Physical activity (PA) is an underutilized preventive strategy that has been shown to reduce the risk of coronary heart disease, cancer, diabetes, arthritis, depressive disorders, and cognitive impairments in older adults [1]. The majority of the costliest chronic conditions among adults 50 years or older can be mitigated or managed with physical activity [2], and significant health benefits can be achieved through even moderate increases in physical activity in older adults aged ≥65 years [3]. Despite the known health outcomes, the Centers for Disease Control and Prevention state that inactivity, or not getting any physical activity beyond basic movement from daily life activities, increases with age to 26.9% in older adults aged 65–74 years and 35.3% in those aged ≥74 years [3], and that inactivity is associated with the most prevalent chronic diseases in this population [3]. In order to encourage healthy aging practices, it is critical to identify modifiable factors that may help increase PA in older adults.

The physical environment has been shown to have an effect on the promotion of PA specifically by increasing walkability, or the capacity of a neighborhood to support individuals’ lifestyle behaviors such as walking and physical activity [4]. While efforts to increase activity, and walkability in particular, are common in urban communities [5], there is limited data related to effective implementation among rural communities. This is problematic as, currently, rural residents exhibit lower levels of physical activity compared with individuals in urban areas [6]. Moreover, this is concerning for the older adult population, as the median age of adults who live in rural areas is 51 years compared with an age of 45 years in urban settings [7]. In 2017, approximately 18% of rural inhabitants were over the age of 65, an increase from 13% in 2009 [8]. As this demographic shift continues, targeted strategies must be identified to create initiatives in the communities where a large portion of older adults live.

Without a robust evidence base for increasing PA in rural communities, translating lessons from urban strategies may be necessary to immediately address the low levels of physical activity in rural-dwelling adults. Modifying effective urban models could demonstrate success in the short term, as older rural adults indeed indicate they perceive the benefits of physical activity, such as improved health, independence, increased mobility, and social cohesion [9], and also prioritize physical activity that is perceived as productive and useful [10]. As walking, gardening, and yard work are some of the most popular forms of physical activity among older adults [3], strategies from successful urban initiatives could be adapted to promote these activities in rural communities.

Rural–urban health disparities are large and can be explained in part by rural and urban built environment differences at the neighborhood level [11]. NHANES (National Health and Nutrition Examination Survey) data demonstrates that the prevalence of overweight and obesity in rural adults is partly attributable to neighborhood built environmental features, including walkability, land use patterns, and spatial park accessibility [11]. Thus, the built environment is likely a specific contributor to greater chronic disease risk for older adults in rural settings.

In January of 2017, the Federal Highway Administration (FHWA) released the “Small Town and Rural Multimodal Networks (STAR Guide)” to help small towns and rural communities support safe, accessible, comfortable, and active travel for people of all ages and abilities. The report noted many hurdles to implementing walking and bicycling interventions in rural communities, which include agricultural operations, public land access, auto-oriented roadways, lack of transportation options, small town centers, and constrained terrains [12]. The lack of proper infrastructure in rural and under-resourced communities may create a particular dilemma for older people because this population is potentially more susceptible to the adverse effects of physically challenging environments, and mobility, independence, and social interaction may be potentially limited by inadequately designed neighborhoods [13]. As such, addressing rural infrastructure and resource limitations may be a key to increasing the walkability of these communities.

In this review, we provide best practices in urban neighborhood initiatives that could potentially be implemented in rural, resource-limited communities to increase PA in older adults. Based upon data drawn from urban models, we identified three low-cost models that can specifically increase walkability in rural communities, including microscale and pop-up infrastructure, municipal parks, and community gardens.

## 2. Materials and Methods

A systematic review method was used that complied with the Preferred Reporting Items for Systematic Review and Meta-Analysis Protocols (PRISMA-P) [14] (Figure 1). PubMed, Medline, Academic Complete Search, and CINAHL Complete were used as the primary research engines for our literature review to identify relevant studies on the relationship between urban neighborhood initiatives and increasing walkability in rural communities for older adults. Inclusion criteria incorporated original research articles published in English between 2009 and 2019 with the main objective being to study the association between an urban neighborhood initiative and increasing walkability for older adults. Studies with a focus on the relationship between neighborhood initiatives and increasing physical activity or health outcomes in older adults were included. Studies that did not analyze the association of the neighborhood built environment and walkability or physical activity were excluded. Studies that only analyzed children, adolescents, or young adults were also excluded. To determine whether initiatives were appropriate to incorporate in a rural setting, the Star Guide recommendations were used [12]. Specifically, initiatives must be a part of a multimodal network that is “[a] safe and direct network providing convenient access to key destinations, while minimizing exposure to motor vehicle traffic… [emphasizing] physical safety [and] user comfort” [12].

Original search terms included physical activity or exercise or fitness or physical exercise, elderly or aged or older or elder or geriatric, walking or walkability or pedestrian friendly, streetscape or pedestrian infrastructure or cycling infrastructure or infrastructure, urban, and rural. After the original search was complete, three members of the research team (A.K., L.V., and R.P.) utilized abstracts to combine similar topics into themes, with at least five articles on each topic needed for further inclusion. Three key themes were identified: microscale and pop-up infrastructure, parks, and community gardens. Search terms were modified to include parks or recreation, physical activity or exercise or fitness or physical exercise, elderly or aged or older or elder or geriatric, urban parks or recreation or green space or parks or natural environment, community gardens or urban gardens, walking or walkability or pedestrian friendly, streetscape or pedestrian infrastructure or cycling infrastructure or infrastructure, urban, and rural. Suitable and complete manuscripts were checked independently by two reviewers (A.K. and L.V.) with regard to the inclusion criteria. In case of any divergence, an additional reviewer was consulted (R.P.), and a consensus was reached through discussion. A total of 143 studies were identified, of which 18 studies met the inclusion criteria and were used in this review. Eleven additional studies were found within the citations and were also included for a total of 29 studies (Table 1).

## 3. Results

### 3.1. Microscale and Pop-Up Infrastructure

Microscale, or neighborhood level, infrastructure includes solutions such a maintained footpaths [35], availability of street lights [35], presence of sidewalks/protected walkways [36,37], quality of street crossings [20], street connectivity [39], recreational facilities [28], as well as rest stops and benches [35]. Microscale changes not only increase walkability [41], but have also been shown to have a positive impact on walking for travel in older adults [20,28,34,35,36,37,40,42]. Microscale infrastructure also positively affects neighborhood aesthetics, which has been shown to increase leisure and utilitarian walking [34,40,42].

Pop-up infrastructure, defined as affordable, flexible, quickly deployable, modular, and relocatable structures, is a feasible way to test out potential infrastructure changes to understand community acceptance and engagement [44]. Pop-up infrastructure has been shown to be an effective strategy to increase walking and overall physical activity [44]. For example, rest stops, including benches or restrooms, have been shown to specifically increase walkability [33,35]. Among urban community members aged 61–89 years, pop-up benches were ranked as a top priority when developing a walkable neighborhood [33]. Moreover, they created many positive mobility experiences by (1) increasing the use and enjoyment of green and blue (river, lake, and ocean front) spaces; (2) assisting as a mobility aid for those with physical limitations or chronic disease management, as participants tended to walk on paths that had benches; and (3) contributing to older adults making and maintaining social connections and improving social capital [33].

While pop-up infrastructure changes are often small scale and short term, larger and more long-lasting initiatives have also been implemented. A successful large pop-up infrastructure in Los Altos, CA, a city located in the Bay Area of San Francisco, demonstrated that a two-year temporary pop-up park increased the frequency of park use and the physical activity level of the park users [25]. Researchers found a greater than 10% increase in moderate-to-vigorous physical activity (MVPA) in a one-year period for park participants [25]. In Cleveland, OH, increasing the greenery and walkability of local streets was associated with a decrease in clinical outcomes such as BMI [31].

Not only have pop-up infrastructure initiatives seen success in older urban communities, but they are already being translated for use in rural communities. The Ottumwa’s Better Block Project in Ottumwa, IA built pop-up benches, pop-up shops, installed curb extensions (also called bumpouts), crosswalks, and temporary bike lanes to make the streets more pedestrian friendly [45]. Residents also increased the aesthetics of the streets and old buildings with public art installations. With the help of 150 volunteers, in under eight hours, “their street turned into a vibrant and lively neighborhood with many attractions for local residents” [45].

### 3.2. Municipal Parks

Municipal parks have been shown to increase walkability and subsequent physical activity in many urban neighborhoods throughout the United States [20,25,27] and internationally [22,23,32]. Not only can establishing a green space in rural neighborhoods increase walkability within the park, but the installation can also become a destination to which to walk or bike. In several urban communities, those who lived closer to any attractive open space participated more regularly in recreational walking and other related activities [24] and the number of parks was significantly and positively associated with increased physical activity [31], including recreational walking [29]. Specifically, older adults who had easy access to parks and recreational facilities were shown to increase walkability and MVPA by 30–59 min a week [24]. Adults with attractive open spaces within 1.6 km of the home were more likely to walk 150 min or more in a week [32]. In older adults who lived alone, recreational walking has been shown to be positively related to the close proximity of parks [24].

Parks can also be modified to include a multitude of walking and physical activity options. This is key, as studies have shown that urban multifunctional parks were used at higher rates than single-purpose parks [26,30]. In Kansas City, MO, the association between park proximity, park facilities, adults’ park use, and park-based PA was examined. Distance to the closest park was not significantly related to either park use or park-based PA; however, multiple diverse facilities were associated with park use and park-based PA. Older adults, specifically, had a higher likelihood of using a park that featured a basketball or tennis court, fitness stations, trails, or a swimming pool [30]. This could be an important element, as sport-related amenities may prolong older adults’ physical fitness and bodily coordination [46].

Besides the potential multifunctionality within a park, diverse facilities around or near the park can also increase walking. In Portland, OR, urban older women residing in close proximity to parks with convenient access to amenities, including transit and commercial areas, were more likely to walk for leisure as well as leisure and utilitarian purposes combined than older women not in close proximity to parks [27]. In Paris, France, it was similarly demonstrated that the density of destinations, including lakes and waterways, was associated with an increase in time spent walking for recreational purpose [26].

Using parks or green spaces as a location for planned physical activity by establishing targeted activities or gatherings for older adults within rural parks, such as physical activity classes, walking groups, gardening, or other means of park utilization, also shows promise. In Brazil, supervised physical activity classes that were held in urban outdoor parks drew more older adults than parks without social programming [21]. A similar program, called Fit and Strong, was offered in 26 states, including in many rural communities. This program targeted older adults with osteoarthritis and provided a physical activity, behavior change, and fall-prevention program. They found that after eight weeks, participants gained confidence in exercising, decreased stiffness, improved joint pain, and improved lower extremity strength and mobility [47].

Park restructuring is already being implemented and examined in rural communities. Albert Lea, MN, a rural community, is applying the Blue Zones wellness framework, which encompasses nine core tenets designed to help people move naturally, eat wisely, connect and create the right outlook, and deepen their sense of purpose, specifically around rural green spaces [48]. Multiple initiatives were implemented, including renovating a park with an amphitheater to increase functionality and social gatherings, creating walking groups, hosting community events, and making physical infrastructure changes [48]. These changes were correlated with an approximately 70% increase in daily step totals and increased the lifespan of the residents by 2.9 years.

### 3.3. Community Gardens

Community gardens have increased in popularity throughout the last century, as they enhance neighborhood walkability [4] and have been associated with improved health behaviors, including increased PA [49] as well as reducing depressive symptoms [50]. Gardens could be a particularly targeted initiative for older adults, as it has been reported that older adults prioritize physical activity perceived as productive and useful [10], where walking, gardening, and yard work are some of the most popular forms of physical activity in older adults [3]. Gardening has also been shown to have an array of positives influences on many aspects of older adult life, including physical, nutritional, social, and psychological outcomes [16,18].

Accessibility is key to increasing gardening in older adults. A secondary analysis of the LIFE-P study, which examined 70–89-year-old adults at risk for mobility disability, determined that those who lived in less compact neighborhoods spent more time performing heavy gardening compared with those in more compact neighborhoods [15]. Individuals who lived in less compact neighborhoods were also more likely to maintain or increase their participation in gardening and yard work as well as increase walking for leisure [15] after a 12-month PA intervention. In Wales, United Kingdom, older men aged 45–69 years reported more physical activity engagement, including walking and gardening, in neighborhoods with more green spaces [51]. Participants aged ≥50 years in Victoria, Australia spent 8–10 h in a community garden, resulting in improved physical fitness. The activity was deemed not too strenuous and was suitable for all older individuals with different fitness levels [16].

Like public parks, community gardens can also help older adults form social networks and support groups. In the Bronx, NY, researchers visited 19 community gardens and interviewed 32 gardeners, and found that 96% of the gardeners reported the benefits of gardening as “staying closer to family”, “neighborhood beautification”, and “family health” [19]. Another study found that participants reported that the community garden provided them with a sense of belonging and purpose, connection to their local community, and accomplishment, as well as encouraged sharing, trust, and friendship [16]. In a nursing home setting, gardening positively impacted the resident’s quality of life with meaningful daily activities, enjoyment of daily life, relationships, and independent living [17].

Some rural examples, such as the Community Work Day in Maui, HI [52] and Community Coalitions for Change in rural Tennessee [53], have shown that the addition of a community garden brought communities together, provided access to more fruits and vegetables, and increased activity levels [52,53]. Although there is a lack of research on rural community gardens for older adults, the addition of gardens in rural communities have the opportunity to not only increase the walkability of the neighborhood but also increase physical activity overall, as well as improve social connectivity for older adults.

## 4. Discussion

As the rate of chronic illnesses and cost of treatment continue to rise with physical inactivity in the United States, there is an immediate demand to promote physical activity by increasing walkability, especially in rural neighborhoods where fewer people engage in leisure-time physical activity and meet the physical activity recommendations [6]. More so, older adults tend to live in rural communities [6], which may exacerbate lower levels of physical activity [3] in this demographic.

In general, it has been considered difficult to promote physical activity in rural settings because of cost concerns and environmental challenges, such as the lower density of development, distance between destinations, and the nature of rural infrastructure (e.g., roads with high-speed traffic, lack of sidewalks, etc.) [11,12]. These challenges are particularly noteworthy for older adults, as their mobility decreases with age and physical barriers become more significant [13]. However, there appear to be opportunities to capitalize on some advantages that rural environments might provide that are generally inexpensive and mirror evidence-based urban initiatives.

Microscale and pop-up infrastructure (pop-up retail, public art, services, and activities) are urban solutions that enhance the public realm and can boost community relationships, culture, and activity, all of which can be anchors of engagement for an elderly rural population. Moreover, as resources are often limited in rural communities, permanent high-cost initiatives could be a generally undesirable first implementation step [12]. Simple solutions such as rest stops, benches, and other moveable pop-up infrastructure to enhance walkability and improve aesthetics could be relatively easy to install and inexpensive, particularly compared with high-burden, long-term, individually targeted interventions. This may be key, as the most important factor in promoting walking in rural towns, according to one study, is aesthetics [43] and relaxing destinations incorporating environmental elements have been associated with increased transportation walking and walking overall [38].

Other important impacts of such interventions are likely the increases in perceived and actual pedestrian safety. A curb extension at a crosswalk, for example, improves a pedestrian’s visibility around parked vehicles, shortens the pedestrian’s crossing distance and therefore time spent in the road, and can slow turning vehicles [12]. Similarly, the addition of a bicycle lane can act as a buffer between a sidewalk and the motor vehicle lane, which can improve pedestrian comfort and sense of safety. This is important to consider, as feeling unsafe influences rural older adults’ walking behaviors [34].

With regard to parks and community garden sites, rural neighborhoods may have more open space in general, which could provide an opportunity to create shared areas in communities that encourage walking and social interaction. In small towns and rural communities, the land outside of the commercial district and downtown area is much less dense and has wider green space in between destinations [47]. As such, it may be easier in rural, rather than in urban, settings to create linear parks, greenways, and trails, as the availability of open space may make it easier to add on to or improve the facilities in already established parks, such as the addition of a walkway or path. These connections could be critical, as poor accessibility has been cited as a major barrier to walking in rural neighborhoods, while conversely, perceived accessibility was positively associated with destination walking [43].

Repurposing already existing green spaces into outdoor recreational spaces and gardens can provide an affordable and effective solution to extend walking and physical activity opportunities for older rural adults specifically [12]. Establishing community gardens could increase walkability, resulting in higher levels of walking and overall physical activity in older adults, specifically by encouraging older adult engagement in productive activities such as gardening and yard work [10]. Multifunctional green spaces can also serve as a setting for social interaction and gatherings, which could be a critical element for rural older adults, as social well-being accounts for one of the biggest differences between rural and urban communities [54]. A higher frequency of social behaviors is associated with higher levels of self-reported walking for transport and recreation by older adults [55], and specifically for rural older adults, having more friends is associated with overall better health [56]. Recreational walking was shown to increase with age in rural neighborhoods and, in the same community, the social environment appeared to be a major driver behind increased recreational walking [42].

## 5. Conclusions

Enhancing walkability in communities has been shown to increase physical activity, decrease comorbidities, and improve health outcomes. This is of particular importance for vulnerable populations such as older adults, who account for a substantial proportion of society. There is evidence that low-cost built environment approaches to increase walkability and decrease physical inactivity have been successful in urban settings, which suggests potential for under-resourced rural environments. While long-term infrastructure solutions are certainly needed, this review outlined the evidence for three urban initiatives that could be implemented quickly and efficiently in rural communities: microscale and pop-up infrastructure, parks, and community gardens. Some local rural initiatives have demonstrated that these initiatives may, in fact, be accepted and could potentially increase physical activity for all community members [10,33,35,42,45,48,57]. However, both short- and long-term intervention and evaluation data in rural communities is severely lacking, particularly with respect to older rural adults, and the long-term impact is yet unknown [58].

As is the case with many public health interventions in their infancy, further research is needed to examine walkability interventions and health outcomes in the rural aging population. Specifically, future longitudinal implementation studies should be multilevel and examine not only changes in walkability but also sociocultural acceptance of changes in the built environment, utilization by community members, and, finally, changes in physical activity markers and related clinical outcomes.

Although understudied in rural communities, microscale and pop-up infrastructure appear to have the potential to improve walkability in urban and rural neighborhoods, with limited cost and resources. While traditionally PA interventions have been added to schools and workplaces [58] and may be reasonable intervention points for rural communities as well because those organizational units exist there too, smaller evidence-based models are becoming an important first step due to the cost burden reduction when compared with structural change [12]. Moreover, interventions at schools and workplaces tend not to target the oldest members of the population, while smaller initiatives could be particularly suited to rural communities with downtown areas that are in need of revitalization as well as more natural green space, and would target all segments of the population, including older adults.

Lastly, more research specifically targeting rural older adults, community connectedness, and physical activity is needed as available evidence is lacking [58], but some existing interventions demonstrate that making open spaces accessible as well as increasing related social opportunities may significantly increase recreational and destination walking [42,55,56]. Barriers such as poor funding and lower frequency of grant success in rural municipalities compared with urban municipalities has been noted [57], and future funding and research should be allocated to address this gap in practice and the literature

Improvements in walkability and successful promotion of physical activity by altering the built environment require collaborative efforts by people across disciplines and at multiple levels of society, from policy through to the individual. While successful urban initiatives can be used as a guide, a systematic implementation and evaluation approach should be undertaken in order to understand the unique implications of altering the built environment in rural communities to make them more walkable, particularly for older adults.

## Figures and Tables

**Figure 1 ijerph-16-03041-f001:**
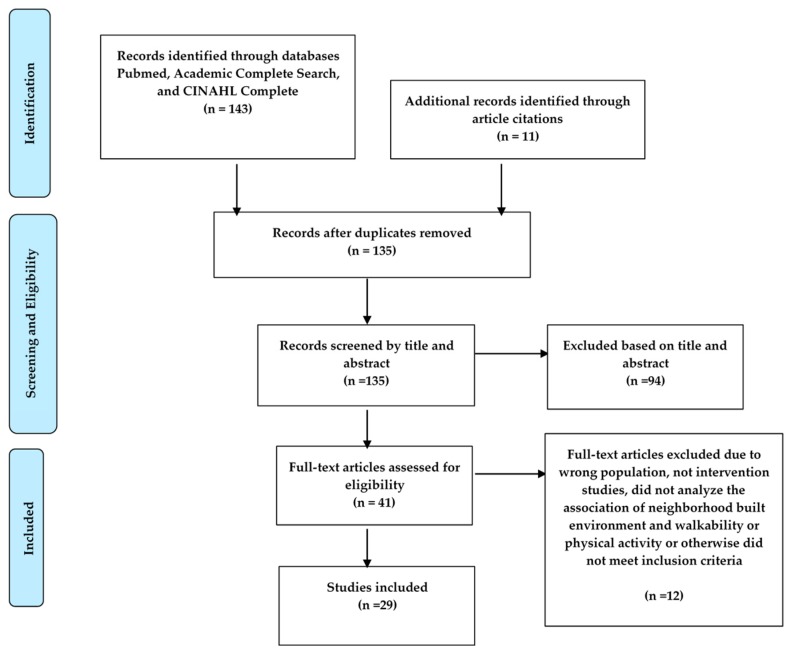
Article Selection Flow Chart.

**Table 1 ijerph-16-03041-t001:** Included Articles.

Author	Year Published	Built Environment Intervention	Purpose	Study Design	Results
King et al. [15]	2017	Community Gardens	Explored whether the compactness of the neighborhood in which a participant lived moderated the effects of the physical activity	Randomized Controlled Trial	Participants that were encouraged to increase PA and were living in less compact neighborhoods, were more likely to maintain or increase their participation in gardening or yard work
Sanchez et al. [16]	2017	Community Gardens	Evaluated the perceived health-related benefits of community gardens for older rural citizens and how might community gardens be improved for the rural community.	Prospective cohort	Findings indicate that there is a range of health-related benefits associated with participation in rural community gardens, including physical, nutritional, social and psychological
Raske et al. [17]	2009	Community Gardens	Evaluated the impact of the construction and use of an enabling garden on resident quality of life in a rural nursing home	Cross Sectional	Findings suggest the garden had positive effects on resident quality of life, particularly in terms of meaningful daily activities, enjoyment of daily life, resident relationships, and functional competency
Weltin et al. [18]	2012	Community Gardens	The purpose was to monitor HbA1c levels in the Marshallese population who participated in a community garden.	Randomized Controlled Trial	Persons who participated in a community garden had significant reduction in their HbA1c postintervention, compared to persons who did not participate actively.
Ottman et al. [19]	2010	Community Gardens	Evaluate quality of life outcomes of individuals participating in a community garden.	Cross sectional	Gardeners stated gardening keeps them closer to the family,” “neighborhood beautification,” and “family health,”. The gardeners’ perception about how the Community Gardens have improved quality-of-life
Todd et al. [20]	2016	Parks	To derive empirically defined and measured built environment (BE) features and individual outcomes using pooled data from a sample of older adults residing in 2 U.S. metropolitan regions.	Cross sectional	The most common profile was labeled "low walkable, low transit access, low recreation access” and Sedentary time significantly differed profiles.
Parra et al. [21]	2010	Parks	To assess park use in Recife, Brazil, and differences in physical activity and occupation rates in public parks with and without cost-free, supervised PA classes.	Cross sectional	Sedentary level were lower in ACP sites compared to non-ACP and more people were likely to be engaged in vigorous PA in ACP sites compared to non-ACP.
Ribeiro et al. [22]	2013	Parks	Examine the relationship between socio-environmental characteristics of neighborhood of residence and the frequency of leisure-time physical activity (LTPA) among older adults from Porto (Portugal).	Cross sectional	A majority of the participants reported no LTPA and neighborhood characteristics were unrelated to whether older people exercised or not.
Cerin et al. [23]	2013	Parks	To identify the aspects of the neighborhood environment associated with LTPA of Chinese elders residing in an ultra-dense city.	Cross sectional	Recreational walking was positively related to the availability of parks and other environmental attributes.
Carlson et al. [24]	2012	Parks	To evaluate ecological model predictions of cross-level interactions among psychosocial and environmental correlates of physical activity in community-dwelling older adults.	Cross sectional	Across all interactions, living in a supportive environment was related to more min/wk of PA.
Salvo et al. [25]	2017	Parks	To document patterns of park use and levels and correlates of park-based PA at a temporary pop-up park implemented in Los Altos, CA	Cross sectional	Most park users were adults, including of older adults and park users were more engaged in moderate to vigorous PA.
Perchoux et al. [26]	2015	Parks	To understand how built environment characteristics influence recreational walking is of the utmost importance to develop population-level strategies to increase levels of PA in a sustainable manner.	Cross sectional	Overall, a high density of destinations, the presence of a lake or waterway, and a high neighborhood education were associated with higher odds of recreational walking
Siu et al. [27]	2012	Parks	The association btw built environments and walking among older women by developing refined built environment measures i order to identify distinct urban forms.	Cross sectional	Older women residing in city center were more likely to walk than those living in city periphery, suburb communities, and urban fringe with poor commercial access.
Barnett et al. [28]	2017	Parks	The aim of this study was to systematically review and quantify findings on built environmental correlates of older adults’ PA and investigate differences by type of PA and environmental attribute measurement.	Systematic review and meta-analysis	Positive environmental correlates of PA, ranked by strength of evidence, were: walkability, safety from crime, overall access to destinations and services, recreational facilities, parks/ public open space and shops/commercial destinations, greenery and aesthetically pleasing scenery, walk-friendly infrastructure, and access to public transport
Chaix et al. [29]	2014	Parks	Examined whether numerous street network-based neighborhood characteristics related to the sociodemographic, physical, service, social-interactional, and symbolic environments were associated with overall recreational walking and recreational walking in one’s residential neighborhood	Cross sectional	A higher neighborhood education, a higher density of destinations, green and open spaces of quality, and the absence of exposure to air traffic were associated with higher odds of recreational walking and/or a higher recreational walking time in one’s residential neighborhood.
Kaczynski et al. [30]	2014	Parks	The purpose of this study was to examine associations between park proximity and park facilities and adults’ park use and park-based PA, while also exploring differences by gender, age, race, and income	Cross sectional	Distance to the closest park was not significantly related to either park use or park-based PA. However, numerous significant associations were found for the relationship of number of parks and amount of park space within 1 mile with both outcomes.
Sallis et al. [31]	2016	Parks	Examined how objectively measured attributes of the urban environment are related to objectively measured physical activity, in an international sample of adults	Cross sectional	The number of parks was significantly and positively associated with physical activity.
Sugiyama et al. [32]	2010	Parks	Examined associations of attractiveness, size, and proximity of multiple neighborhood open spaces with recreational walking.	Cross Sectional	Shorter distance to attractive open spaces was associated with doing any recreational walking, but adults with larger attractive open spaces within 1.6 km of their home were more likely to walk 150 minutes or more in a week.
Ottoni et al. [33]	2016	Microscale and Pop-up	Examined how one microscale feature (benches) influence older adults experiences of mobility and well-being.	Qualitative	Benches positively contributed to older adult’s mobility experiences by (1) enhancing their use and enjoyment of green and blue spaces (water: lakes, rivers, oceans, and ponds). (2) serving as a mobility aid (3) contributing to social cohesion and social capital.
Van Cauwenberg [34]	2012	Microscale and Pop-up	To examine the relationship between residential areas (urban, semi-urban, and rural) and walking and cycling for transportation and recreation in Flemish older adults.	Cross Sectional	Urban older adults were more likely to walk for daily transportation compared to semi-urban and rural older adults. Semi-urban older adults were more likely to bike for daily for daily transportation compared to urban and rural older adults. Accessibility and feelings of safety were essential to promote active commuting.
Cerin et al. [35]	2017	Microscale and Pop-up	To study the correlation between active traveling in older adults and neighborhood physical environments.	Systematic review & meta analysis	Littering/vandalism/decay was negatively related to total walking for transport. Positive associations were observed with food outlets, business/institutional/industrial destinations, availability of street lights, easy access to building entrance and human and motorized traffic volume.
Forjou et al. [36]	2017	Microscale and Pop-up	This study examined the association between selected sociodemographic, health, and built environmental factors and walking behaviors of middle-aged and older overweight/obese adults	Cross sectional	Walking the recommended ≥150 min per week for any purpose was significantly associated with having at least a college degree, having no difficulty walking a quarter of a mile, and being unemployed as well as perceived presence of sidewalks/protected walkways and perceived absence of distracted drivers in the neighborhood.
Mäki-Opas et al. [37]	2016	Microscale and Pop-up	This study examines whether the proximity of green space and people’s residence in different travel-related urban zones contributes to commuting physical activity	Cross sectional	Higher levels of commuting physical activity were associated with pedestrians who lived in a main centre or sub-centre. Women who lived near to a public transportation were twice as likely to be physically active while commuting compared to men.
Cain et al. [38]	2014	Microscale and Pop-up	To examine the associations between microscale infrastructure and multiple physical activity across four different age groups (children, adolescents, adults, older adults).	Cross sectional	Among all age groups, Microscale Audit of Pedestrian Streetscapes (MAPS) scores were significantly associated with walking/biking for transport, leisure/neighborhood PA, and objectively-measured PA.
Jack et al. [39]	2014	Microscale and Pop-up	To examine if built environment moderates the association between self-reported measures of the neighborhood built environment and walking.	Cross Sectional	LW neighborhoods, respondents in HW neighborhoods positively perceived access to services, street connectivity, pedestrian infrastructure, and utilitarian and recreation destination mix, but negatively perceived motor vehicle traffic and crime related safety.
Troped et al. [40]	2016	Microscale and Pop-up	To examine the associations between neighborhood built environment with leisure, utilitarian walking, and meditation by the perceived environment among older women.	Cross Sectional	Perceived land use mix and aesthetics significantly predicted leisure and utilitarian walking,
Li et al. [41]	2018	Microscale and Pop-up	To determine the influence of street greenery and walkability on body mass index in Cleveland, Ohio, USA.	Cross sectional	Walk Score has a more significant association with decreased BMI for males than females and the street greenery has a more significant association with decreased BMI for females than males in Cleveland, Ohio.
Li et al. [42]	2018	General Walking Behavior	Examined the perceived neighborhood characteristics and environmental barriers in association with two different types of walking - recreational and destination - in the context of a rural town	Cross sectional	Perceived aesthetics were significantly associated with more frequent recreational and destination walking. Higher perceived accessibility were associated with more frequent destination walking.
Whitfield et al. [43]	2019	General Walking Behavior	To identify the significant associations between supports and destinations with walking among a nationally-representative sample of urban- and rural-dwelling adults	Cross sectional	Among all participants, roads, sidewalks, paths, or trails and relaxing destinations were associated with leisure walking. Among urban residents, sidewalks on most streets and all four destination types were associated with transportation walking; among rural residents, roads, sidewalks, paths, or trails; movies, libraries, or churches; and relaxing destinations were associated with transportation walking.

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
