# Peer review of "Translating Urban Walkability Initiatives for Older Adults in Rural and Under-Resourced Communities"

_ijerph, 2019, doi:10.3390/ijerph16173041_

Round 1
Reviewer 1 Report
Thank you for the opportunity to review this revised manuscript. The authors have adequately addressed the previous concerns of this reviewer, and there are only minor comments below to address:
In the second paragraph of the introduction, you mention the demographic shift in rural areas, but two paragraphs later, you discuss education, income and environment as factors related to health. Since the rural population is experiencing increases in the aging population, and you do not really address the income and education issue, you can probably remove the discussion on of education and income.
I did not see the table or figure attached and would like to review them.
Author Response
Reviewer 1 Comments | Revisions |
In the second paragraph of the introduction, you mention the demographic shift in rural areas, but two paragraphs later, you discuss education, income and environment as factors related to health. Since the rural population is experiencing increases in the aging population, and you do not really address the income and education issue, you can probably remove the discussion on of education and income. | The mention of income and education has been removed from this section. |
I did not see the table or figure attached and would like to review them. | We apologize this was not included in the previous upload and have uploaded again. |
Regarding the limits (last sentence on page 2), the authors could rephrase theirs as "studies published between 2009-2019", and "adults aged 65+ years". Also, it might be notable to mention that 2009-2019 actually covers 11 years of publications. | These phrases have been updated. Given that publications were only includedbetween Feb 2009 and Feb 2019, there are only 10 years of publications. |
The first sentence in the 2nd paragraph on page 3 can be removed - it is a little redundant given the edits in the preceding paragraph. If that sentence is deleted, the remaining portion of that paragraph can be combined with the previous one. | Micro-scale and pop-up infrastructure are two related, but distinctly different approaches to built environment changes. Micro-scale are essentially small projects, while pop-ups are inherently modular. We were asked by reviewers in the last revisions to make this differentiation clear and believe it is important to keep them in separate paragraphs. |
It is worth noting that the authors did not define 'sedentary behavior", which is not the same thing as physical inactivity. | This is a key distinction and oversight. Our focus is not on sedentary behavior. As such, we have changed the three times “sedentary behavior” was mentioned to “physical inactivity” as it was referred to in the original manuscripts. This eliminated the need to define sedentary behavior, as it is no longer included in our discussion. |
Per PRISMA guidelines, it would be helpful if the data items or summary measures were discussed. | While we followed the PRISMA guidelines for this literature review, we did not perform any statistical analysis, so a summary measure would not be appropriate. However, we have highlighted all included articles in Table 1, where outcomes have been noted. |
Reviewer 2 Report
The aim of the paper is interesting, however there are some imprecisions and missings on the scientific method used to review and to analyse the selected papers.
Here my comments:
- p. 2 : how did you identify the three themes? Why these three and not others?
- p.3: which are the inclusion criteria? They are linked only to the keywords and “limits” or on the topic, method, aim of the papers?
- How the literature deals with the problem of increasing PA in rural communities? There are differences among nations, or among research field? There are differences among the method used to design such practices? How the authors analyse these aspects in the review is not clear
- How the authors select best practices in urban context? Why these three themes are the most suitable? What are the links among such best practices in urban context and in rural context? Authors should define the criteria and the method used to select such best practices and why these are suitable for rural context and for older adults and PA.
- a table with the analysis of the selected papers should be added.
Author Response
Reviewer 2 Comments | Revisions |
p. 2 : how did you identify the three themes? Why these three and not others? | A sentence has been included to identify the two step process for how themes were identified. |
p.3: which are the inclusion criteria? They are linked only to the keywords and “limits” or on the topic, method, aim of the papers? | Inclusion/exclusion criteria and search terms have been made more clear in the methods section. |
How the literature deals with the problem of increasing PA in rural communities? There are differences among nations, or among research field? There are differences among the method used to design such practices? How the authors analyse these aspects in the review is not clear | We have included a new citation [61] as well as added a paragraph to the discussion to address this concern. However, the major gap we are trying to fill with this literature review is the absence of research in rural communities, particularly among older adults. As such, an in depth comparison of interventions between fields and countries is slightly premature, as this data is severely lacking. |
How the authors select best practices in urban context? Why these three themes are the most suitable? What are the links among such best practices in urban context and in rural context? Authors should define the criteria and the method used to select such best practices and why these are suitable for rural context and for older adults and PA. | This has been addressed in the methods section and a definition by the STAR Guide has been included for clarification. |
a table with the analysis of the selected papers should be added. | We apologize this was not included in the previous upload and have uploaded again. |
Round 2
Reviewer 2 Report
Thank you for your email. For me there is not other issue for publication.
This manuscript is a resubmission of an earlier submission. The following is a list of the peer review reports and author responses from that submission.
Round 1
Reviewer 1 Report
Thank you for the opportunity to review this manuscript, which provides an important review of the issues surrounding older adult physical activity. The authors provide a good description of the problem of low activity in this population, and the review provides a decent look at what has been done. Below are some comments to address:
One general comment throughout: physical activity, physical inactivity, sedentary behavior, and walkability are all separate terms, and in particular, physical activity and sedentary behavior are independent risk factors for diseases. Please be careful with your use of these terms throughout, and be consistent with the most relevant terminology for this manuscript.
Abstract: Maybe provide some brief examples of the built environment
Line 33: please define inactivity
Lines 37-39 need citations
Line 58 – Could you please cite, and briefly describe these rural-urban educational/environmental differences
Line 62- Insert “guide” after (STAR)
Line 70 – This could use a citation
Line 73 – instead of saying “can be”, you might perhaps state “could potentially be”?
Materials and Methods: Need more detail here - Who reviewed the studies to reduce 143 to 18? How were disagreements handled? Was this review guided by PRISMA, or another framework? Could you provide a flowchart of the articles found and included? How are you defining older adults (i.e. what is the age range)? If needed, see the PRISMA guidelines (http://www.prisma-statement.org/) for what to include. Further, can you speak to how encompassing this review was – for example, with no limiters, a Pubmed search for “physical activity in older adults” brings up 25,537 items. Besides using the time frame of the past 10 years, how was this many studies, with your variety of search terms, narrowed down to 143, then 18?
Results: Could you make a summary table of included studies? Also, were there really only three categories of strategies (pop-up infrastructure, parks, and gardens)? Were no other categories found in the data (e.g. active transportation, workplace activity, etc.)?
Line 161: MI should be MO for Missouri
Line 168: Can you describe what “making sense of aging” means, and how it is related to physical activity? Not sure how it fits into this manuscript. Maybe delete it.
Line 182 – you might mention something about the Blue Zones for readers unaware of what this is
Discussion: Overall, the discussion cold be expanded - think about what some of the literature shows related to PA in rural adults - there are plenty of studies to discuss.
I know that barriers are not the focus of this review, but perhaps if space permits, you could address or expand on some of the rural barriers or challenges to implementing the three types of strategies in your results section. For instance, the density of destinations (line 174) may be difficult to reproduce in rural areas due to a variety of reasons (no funding, no resources, no interest, etc.)
Lines 258-68: Any of these sentences could use citations and connections with your findings/the literature.
Conclusions: This first sentence can be deleted as this review is more focused on activity and walkability. Also, your conclusions could be tied to your results better – of the three strategies in the results, which has the most promise? Which, if any, have evidence of success in rural areas? You might mention future directions that will directly result from this review.
Reviewer 2 Report
Major comments:
A better explanation of the search strategy and study eligibility criteria and study selection should be included in the methods section (e.g., rural does not appear as a search term).
How many studies were excluded? How did the authors assess if the studies had a rural focus in order to be included?
Authors should include a Figure illustrating the study flow. Moreover, there is a need for Table 1 summarizing the included studies.
Further, there is a lack of information about the search procedures and the data extraction process in this methods section (e.g., who conducted the data extraction? Who selected the included articles? Did pair of reviewers conduct the selection and data extraction?)
Authors need to specify the definition of the main variables studied.
Minor comments:
Introduction
Line 37: “The physical environment has been shown to have an effect on the promotion of PA and a growing number of studies have demonstrated this association in older adults”. This sentence needs to include references to these studies
Reviewer 3 Report
The paper aims to review walkability papers in order to examine best practices from urban initiatives that can be implemented in rural communities for older adults. The purpose is interesting but the authors don’t develop a review paper:
- The “materials and method” section is poor and need to be deepen. For example: there is not a summary table of the final set of papers; there is not a critical review of the selected papers; how the authors analyse selected studies? how the study are geographically distributed? areas of research? How the studies are distributed among the years? How they develop urban initiatives for walking?
- The “result” section is interesting but incomplete and the three criteria of analysis are arbitrarily defined. Why this classification (micro-scale; municipal parks; community gardens)? it is possible to have a table or a scheme that explain and synthesize all the method used and proposed by the walkability studies?
- The “discussion” is just a synthesis of the “result” section
- Conclusions are almost non-existent: lessons learned? Future studies?